# Top-Down Proteomics Detection of Potential Salivary Biomarkers for Autoimmune Liver Diseases Classification

**DOI:** 10.3390/ijms24020959

**Published:** 2023-01-04

**Authors:** Alessandra Olianas, Giulia Guadalupi, Tiziana Cabras, Cristina Contini, Simone Serrao, Federica Iavarone, Massimo Castagnola, Irene Messana, Simona Onali, Luchino Chessa, Giacomo Diaz, Barbara Manconi

**Affiliations:** 1Dipartimento di Scienze della Vita e dell’Ambiente, Università di Cagliari, 09042 Cagliari, Italy; 2Fondazione Policlinico Universitario “A. Gemelli”—IRCCS, 00168 Rome, Italy; 3Laboratorio di Proteomica, Centro Europeo di Ricerca sul Cervello, IRCCS Fondazione Santa Lucia, 00179 Rome, Italy; 4Istituto di Scienze e Tecnologie Chimiche “Giulio Natta”, Consiglio Nazionale delle Ricerche, 00168 Rome, Italy; 5Liver Unit, University Hospital of Cagliari, 09042 Cagliari, Italy; 6Dipartimento di Scienze Mediche e Sanità Pubblica, Università di Cagliari, 09042 Cagliari, Italy; 7Dipartimento di Scienze Biomediche, Università di Cagliari, 09042 Cagliari, Italy

**Keywords:** autoimmune hepatitis, biomarkers, primary biliary cholangitis, RF analysis, salivary proteomics, top-down proteomics

## Abstract

(1) Autoimmune hepatitis (AIH) and primary biliary cholangitis (PBC) are autoimmune liver diseases characterized by chronic hepatic inflammation and progressive liver fibrosis. The possible use of saliva as a diagnostic tool has been explored in several oral and systemic diseases. The use of proteomics for personalized medicine is a rapidly emerging field. (2) Salivary proteomic data of 36 healthy controls (HCs), 36 AIH and 36 PBC patients, obtained by liquid chromatography/mass spectrometry top-down pipeline, were analyzed by multiple Mann—Whitney test, Kendall correlation, Random Forest (RF) analysis and Linear Discriminant Analysis (LDA); (3) Mann—Whitney tests provided indications on the panel of differentially expressed salivary proteins and peptides, namely cystatin A, statherin, histatin 3, histatin 5 and histatin 6, which were elevated in AIH patients with respect to both HCs and PBC patients, while S100A12, S100A9 short, cystatin S1, S2, SN and C showed varied levels in PBC with respect to HCs and/or AIH patients. RF analysis evidenced a panel of salivary proteins/peptides able to classify with good accuracy PBC vs. HCs (83.3%), AIH vs. HCs (79.9%) and PBC vs. AIH (80.2%); (4) RF appears to be an attractive machine-learning tool suited for classification of AIH and PBC based on their different salivary proteomic profiles.

## 1. Introduction

Autoimmune liver diseases (AILDs) include a series of pathological conditions that target the liver and have a wide spectrum of presentation, ranging from asymptomatic forms to end stage liver disease requiring liver transplantation.

Despite progress in understanding the etiopathogenesis, diagnostic and therapeutic approach of AILDs, there are still critical issues concerning early diagnosis, risk stratification of disease progression and identification of response to therapy predictors. Indeed, there are several confounding factors involved in the initiation of hepatic autoimmune and inflammatory phenomena such as genetic predisposition, molecular mimicry and/or abnormalities of T-regulatory lymphocytes, autoantibody variability and similarities and overlap syndrome among the three main types of autoimmune liver diseases: autoimmune hepatitis (AIH), primary biliary cholangitis (PBC), and primary sclerosing cholangitis (PSC) [1].

AIH affects females about 3–4 times more than men and afflicts children and adults of all ethnicities and races [2,3]. AIH patients are characterized by a serological increase in antibody and immunoglobulin titers, associated with hepatocellular inflammation, necrosis and fibrosis with possible evolution into cirrhosis and liver failure. AIH patients are distinguished into type 1 (AIH1) and type 2 (AIH2), classified according to autoantibodies profile: smooth muscle antibodies (SMAs) and antinuclear antibodies (ANAs) for AIH1, which occurs more often in adults, whereas liver-kidney-microsome-1 (LKM-1) antibodies characterize AIH2, which develops its most aggressive forms in childhood [4]. 

Management frequently consists of lifelong nonspecific immunosuppression with azathioprine or other salvage therapies [5,6]. 

PBC usually affects women aged 40 to 60, with a male to female ratio of 1:10. It is characterized by the attack of the immune system towards the bile ducts and their epithelial cells, leading to progressive destruction of the intrahepatic bile ducts. Consequently, since the ducts are no longer efficient at draining bile, it accumulates in the liver causing cholestasis and, if the disease is not diagnosed and adequately treated, it develops toward fibrosis, cirrhosis and finally liver failure [7]. Management consists of lifelong administration of ursodeoxycholic acid (UDCA) [8].

PBC is never acute and affects the small bile ducts only, while PSC can affect both small and large ducts [9].

Coexistence of comorbidities is a common finding in AILD patients and may affect clinical phenotype at presentation. The most common association was found with other concurrent extrahepatic autoimmune disorders (CEHAID), mainly with autoimmune thyroid disease, but also with Sjögren’s syndrome, rheumatoid arthritis, nondestructive polyarthropathy, type 1 diabetes, vitiligo, ulcerative colitis and psoriasis [10,11]. Furthermore, an interesting review article reported that celiac disease, which frequently coexists with AILDs, could be genetically linked because both disorders express selected combinations of genes coding for class II HLA molecules on chromosome 6 [12]. The diagnosis and management of AILDs could be challenging in presence of CEHAID since they may predate, coincide or even occur years after the diagnosis of AILDs. To improve AILDs diagnosis, this association has been recognized and incorporated into the original and revised International Autoimmune Hepatitis Group scoring system [3]. Another challenge in AILDs diagnosis is represented by “overlap syndromes” which refer to autoimmune liver diseases that may have characteristics of cholestasis (PBC or PSC) in combination with AIH and that cannot be assimilated into classical diagnostic categories. The Paris criteria can aid in diagnosing the AIH–PBC overlap syndrome [13], while so far, there are no standardized diagnostic criteria for the other types of overlapping [14].

Current guidelines recommend liver biopsy as a prerequisite for the diagnosis, in order to determine disease severity and to discriminate acute and chronic forms of AIH, while recommending against liver biopsy for the diagnosis of PBC, unless PBC-specific antibodies are absent or coexisting with AIH or other systemic co-morbidities [7,15].

To overcome the invasiveness of liver biopsy, peripheral biomarkers on other tissues and biofluids, like blood cells, plasma, eyes and skin have been studied [16,17,18,19,20]. 

Serum autoantibodies ANA/SMA and LKM_-1_ are the diagnostic hallmark of AIH because of their high specificity [3]; however, a recent review reported a very poor diagnostic accuracy for ANA, SMA and LKM_-1_ if detected alone, with accuracy increasing only in presence of both ANA and SMA [21]. Moreover, ANA tests also detect antigen specificities associated exclusively with PBC, including autoantibodies to Sp100 containing nuclear bodies (NBs) or gp210 protein [22]. Among candidate autoantibodies that may aid in the diagnosis of AIH, the most promising is α-actinin, a ubiquitous cytoskeletal cross-linking protein within the family of filamentous actin (F-actin) [23]. Nevertheless, a minority of patients with AIH do not have detectable autoantibodies at presentation and may express them only intermittently or after empiric therapy [3]. As result, such patients must be scored using revised diagnostic criteria (RDC) or liver biopsy. Regarding PBC, although the high specificity of anti-gp210 is generally accepted, the diagnostic role of anti-Sp100 is still questioned since positivity was also found in other liver diseases or immunological disorders without liver involvement [24].

In this scenario, while the possible use of saliva as a diagnostic fluid has been largely investigated for oral and systemic diseases [25,26], it has been only marginally used in autoimmune liver diseases, mainly to investigate the role played by oral microbiota in their pathogenesis [27]. As a mirror of oral and systemic health, saliva provides valuable information because it contains not only proteins specifically secreted by the salivary glands [28], but also proteins from the gingival crevicular fluid [29,30], from oral microflora [31] and plasmatic proteins transported from blood to saliva by both intra- and extracellular pathways. Several studies evidenced that various systemic disorders affected qualitatively and quantitatively the salivary proteome [32,33,34]. 

Emerging omics technologies, together with the sophisticated development of artificial intelligence algorithms, can now accelerate biomarker discovery leading to the use of proteomics for personalized medicine in a rapidly emerging field. In recent years, substantial clinical breakthroughs using Machine Learning (ML) applications have been made, including disease prevention, diagnosis, prognosis, drug discovery and clinical trial design [35,36]. There are multiple examples in the literature where predictive ML models have been used to identify diagnostic biomarkers in immune mediated inflammatory diseases [35,37], liver diseases comprised [38,39,40].

Based on these considerations, the aim of the present study was to evidence by a top-down proteomic pipeline possible qualitative and/ or quantitative differences of targeted salivary proteins/peptides in patients with either AIH or PBC compared with healthy controls (HCs). Mass Spectrometry (MS) data were analyzed by exact Mann—Whitney and Kruskal—Wallis tests. Random forest (RF), one of the most widely used supervised machine learning algorithms for mass spectrometry data, multidimensional scaling (MDS) and linear discriminant analysis (LDA) were used to individuate plausible salivary biomarkers related to AIH or PBC and to accurately classify the subjects.

## 2. Results

### 2.1. Top-Down Mass Spectrometry Pipeline

In this study, we analyzed the most common salivary peptides and proteins soluble in acidic conditions and detectable by RP-HPLC-ESI-MS using a top-down pipeline. The investigated proteins/peptides belong to the following families: acidic proline-rich proteins (aPRPs); statherin and P-B peptide; histatins (Hst); salivary cystatins (S-type); cystatins A, B, C, and D; α-defensins; antileukoproteinase (SLPI); S100A7, S100A8, S100A9, S100A12 proteins. Several variants and post-translationally modified proteoforms, previously characterized in human saliva by our proteomic approach [39], were also investigated. The post-translation modifications (PTMs) considered were phosphorylation, proteolysis, N-terminal acetylation, methionine or tryptophan oxidation, and cysteine S-modification (glutathionylation, cysteinylation, nitrosylation and formation of dimers by disulphide bridges). The UniProt-KB code, experimental and theoretical average mass values (Mav), elution times of proteins and peptides analyzed, *m/z* values and charge of the multiple-charged ions selected for eXtracted Ion Current (XIC) searches in HPLC-low resolution MS and PTMs are reported in Appendix A. All the proteins/peptides listed in Appendix A have been previously characterized by high-resolution tandem MS analysis [41,42]. 

Figure 1 shows the typical total ion current (TIC) chromatographic profile of the acidic-soluble fraction of adult human saliva, and the elution ranges of the protein families investigated in the two patient groups (AIH and PBC patients) and in HCs. Label-free quantification was based on the eXtracted Ion Current (XIC) peak area values measured at low-resolution MS for each peptide/protein in every sample normalized to total protein concentration (TPC). The normalized XIC peak areas values were submitted to the following statistical analyses: (1) non-parametric Mann—Whitney and Kruskal—Wallis tests to identify proteins/peptides with different abundance between groups, (2) Kendall correlations to identify proteins with correlated levels within groups, and (3) random forest (RF) and linear discriminant analysis (LDA) to classify each subject as HCs, AIH or PBC patient. 

Moreover, to increase the functional characterization of varied proteins among the three groups, a gene ontology (GO) enrichment analysis based on biological process (BP) was performed.

### 2.2. Characteristics of the Participants and Saliva Sampling

AIH patients had a median age of 54.53 years and 89% were females, PBC patients had a median age of 65.1 years and 97% were females and HCs had a median age of 50 years and 80% were females. The detailed demographic data of HCs, AIH and PBC patients (in the following indicated as AIHp and PBCp) are reported in Appendix A. Demographic characteristics, including age and gender were matched between the AIHp and HCs and between PBCp and HCs (*p* > 0.05). Clinical and pharmacological features of AIHp and PBCp, collected at the same time of saliva sampling, are reported in Table 1. They comprise a panel of serum markers indices of liver dysfunction and were: markers of hepatocellular dysfunction alanine transaminase (ALT) and/or aspartate aminotransferase (AST); markers of biliary disease such as alkaline phosphatase (ALP); markers of parenchymal liver dysfunction or biliary obstruction, or total bilirubin (TB) and γ-glutamyltransferase (GGT) and albumin useful in assessing hepatic synthetic function. Patients were also tested for antinuclear antibodies (ANAs), smooth muscle antibodies (SMAs) and renal microsomal antigen antibodies (LKMs).

Regarding pharmacological therapies, 41% of AIHp were under both Azathioprine and Steroids, 25.5% under Steroid only, 17.6% under Azathioprine only and 5.5% were without therapy (naïve); PBCp were 100% under UDCA treatment. 

The presence of other concurrent autoimmune diseases was investigated in patients; nine AIHp and nine PBCp presented Hashimoto’s thyroiditis, two AIHp and two PBCp presented rheumatoid arthritis.

### 2.3. Statistical Analysis of the Protein/Peptide Abundances between Groups

Statistical analysis considered both single proteoforms and the sum of different proteoforms of the same protein, which are defined “components” in the text. 

Mann—Whitney and Kruskal—Wallis comparison results of all 71 components measured in HCs, AIHp, and PBCp are shown in Table 2. In addition, the XIC peak areas (25th percentile, median and interquartile range) and the frequencies, are shown in Appendix A. In the case of aPRPs, statherin, P-B peptide, histatins, cystatin A, B, S1, S2, SN, S100A8, S100A9 and α-defensins the sum of the XIC peak areas of all their proteoforms was also considered and reported in Table 2 and Appendix A.

From Table 2 it can be observed that some components showed significant higher levels in AIHp with respect to HCs. In PBCp, some components also showed a significant different level with respect to HCs, some of which with significant statistical differences (*p* < 0.0001).

Results of Mann—Whitney and Kruskal—Wallis tests highlighted that: (i) levels of cystatins S1, S2, SN (components 29–31) and that of the sum of their related proteoforms (components 37–40) were higher in PBCp with respect to both AIHp and HCs; (ii) levels of cystatin A (component 17) and that of the sum of its related proteoforms (component 20) were higher in AIHp, with respect to both PBCp and HCs and with respect to HCs, respectively; (iii) levels of histatin 3, 5 and 6 (components 45–49) were higher in AIHp with respect to both PBCp and HCs, while level of component 49 was higher in AIHp with respect to HCs; (iv) levels of statherin (components 66–67) and that of the sum of its related proteoforms (component 69) were higher in AIHp with respect to both PBCp and HCs; (v) levels of S100A9 proteoforms (components 5, 11, 12) and PRP3_1P (component 61) were lower in PBCp with respect to HCs. It should be outlined that the relative quantification was performed using the XIC peak areas of proteins/peptides normalized by TPC, and that the same results (not shown) were obtained without normalization. Thus, the normalization by TPC, even if affected by abundant proteins not considered in the analysis, did not produce an underestimation of the investigated proteins, and did not introduce errors in the statistical analysis.

### 2.4. Correlation between Protein Levels within Groups

A diagram of correlated proteins/peptides within each group was obtained by multidimensional scaling (MDS) applied to Kendall correlations (Figure 2). To facilitate the understanding of MDS diagrams, the 71 components were subdivided into 12 categories based on their structural/functional similitudes and secretory origin. In MDS diagrams, the closeness between two components denotes the degree of correlation between their concentrations. Thus, MDS clusters represent groups of components that are all strongly correlated with each other.

The most compact cluster, more in HCs than in AIHp and PBCp groups, was that of type 2 cystatins (purple category in Figure 2). In all groups component 26 (cystatin D), 34 (cystatin S1 oxidated), 35 (cystatin S2 oxidated) and 36 (cystatin SN oxidated) showed a tendency to cluster with the category of histatins (yellow category), statherins (blue category) and to a less extent PRPs (green category). All histatins (yellow category) formed a relatively compact cluster in PBCp group, while component 47 (histatin-3) in HCs and components 47 and 43 in AIHp (histatins 3 and 6) were less associated with the main group. Loosely compact clusters, without appreciable differences among groups, were formed by type 1 cystatins (red category) and by S100A9 proteoforms (pink category). However, in HCs, components 4, 8, 9 and 12 of S100A9 proteoforms showed a strong association with defensins (components 50–54, light blue category) while in AIHp and PBCp the same components associated with S100A8 (component 2). 

It is worth noting that PB peptide (single component of black category), which share similar properties with respect to statherin, can be found correlated with aPRPs in all three groups.

### 2.5. Random Forest (RF) Analysis

Confusion matrices of RF classifications of the three mixed data sets, validated by out-of-bag samples, are shown in Figure 3a. RF was applied to a subset of components selected according to the Boruta method (Appendix A). This subset, mainly composed by type 2 cystatins (namely S, S1, S2, SN, A, C and D), PRP3 and histatins, provided a consistent increase of classification accuracy with respect to those obtained using the entire set of proteins/peptides analyzed. Almost all components that showed significant changes with *p*-values < 0.01 or less in Mann—Whitney comparisons, were also selected by the Boruta algorithm for RF analysis, with the single exception of PRP1_3P in HCs vs. PBCp. On the other hand, several components selected for RF, even with high Boruta scores, did not reveal significant changes in Mann—Whitney comparisons. This was especially evident in seven components of the S100A9 category and in S100A7D27, in the comparison between HCs and AIHp. Such discrepancy can be explained by the way decision trees work, as they are able to sort groups by identifying more than one split point in each variable. This means that components with similar means or medians, that do not differ when compared by the Student’s *t* or Mann—Whitney tests, may conversely contribute to discriminate groups based on other differences in data distributions. Classification of samples of the HCs-PBCp mixed data set showed the highest accuracy (83.3%), followed the AIHp-PBCp (80.6%), and HCs-AIHp (79.2%) mixed data sets. Diagrams of RF classifications were obtained by MDS, using the proximity between each pair of samples as distance (Figure 3b). A 3D movie showing MDS of all three groups together is reported in the Appendix A. 

### 2.6. Linear Discriminant Analysis (LDA)

The classification of HCs, AIHp and PBCp subjects obtained by LDA is reported in Figure 4. The overall accuracy of the original classification, considering both true positives and true negatives, was 93% and after cross-validation 82%. Of the 71 components used for LDA, the most discriminant among of the three groups of HCs, PBCp and AIHp were type 2 cystatins (namely cystatins S1, S2, A, B, C and S) and some phosphorylated forms of PRP3. These results are in close agreement with those obtained by RF.

### 2.7. Enrichment Analyses

The results of the GO enrichment analysis of biological process, performed on selected protein with higher Boruta score, are shown in Table 3. The table reports the most overrepresented terms for AIHp and PBCp with respect to HCs. In the case of the HCs-AIHp mixed data set, we found an enrichment of terms mainly related to antimicrobial response mediated by antimicrobial peptide, antimicrobial humoral response and regulation of peptidase and/or protease activity. In the case of HCs-PBCp mixed data set, the enriched terms referred mainly to regulation of proteolysis and negative regulation of peptidase and/or proteases. However, 4 of the 10 biological processes (regulation of peptidase activity, regulation of endopeptidase activity, regulation of proteolysis and regulation of hydrolase activity) were in common in the enriched biological processes of both data sets.

## 3. Discussion

The top-down proteomic approach used in this study allowed the evidencing of quantitative/qualitative differences of naturally occurring selected salivary proteins/peptides among patients with either AIH or PBC and HCs. To our knowledge, this is the first study reporting the feasibility of the salivary proteome for identification of plausible biomarkers of autoimmune liver diseases and for an accurate classification of subjects based on AIH or PBC occurrence. Results obtained by analyzing the proteomic data sets through different statistical approaches are the following: (i) cystatin A, statherin, histatin 3, histatin 5 and histatin 6 were identified as potential salivary biomarkers in AIHp; (ii) proteins S100A12, S100A9 short, cystatin S1, S2, SN and C were identified as potential salivary biomarkers of PBCp; (iii) selected salivary proteins can be used to discriminate AIH and PBC subjects from HCs with relatively good accuracy; (iv) proteins mostly discriminating AIHp from HCs are involved in antimicrobial defense, while peptides/proteins involved in innate immune system characterized PBCp with respect to HCs.

### 3.1. Potential Salivary Biomarkers in AIHp

Mann—Whitney tests provided indications of proteins and peptides on the panel, either secreted or not secreted by the salivary glands, which were elevated in AIHp with respect to both HCs and PBCp. Among these, the most compact group pertains to the histatins family. Members of histatins family are peptides with a high number of histidine residues [43] arising from two parent peptides, histatin-1 and histatin-3. Histatin-1 can be found in either phosphorylated and poly-sulfated forms [44] while histatin-3, due to the presence of a convertase consensus sequence, undergoes a sequential cleavage generation at first histatin-6 (histatin Fr.1–25) and subsequently histatin-5 (histatin Fr. 1–24) [45].

Major interest in histatins originates from the fact that they exhibit inhibitory activities against a broad range of pathogens such as bacteria, fungi, viruses and parasites [46]. The antimicrobial potential of histatin peptides, mainly His-3 and His-5, is also due to the presence of a copper and nickel amino terminal binding site, known as “ATCUN motif” [47,48] which is responsible for the generation of reactive oxygen species that damage membranes of cell organelles and DNA and hence lead to the fungal and bacterial cell death [49,50]. Considering the activity and the relative high abundance of histatins in parotid and submandibular/sublingual secretions [51], these proteins may represent major components of the nonimmune innate host defense system involved in the maintenance of oral health. Indeed, the observed increased level of His-3 and its derived fragments His-5 and His-6 in AIHp, could be linked to oral dysbiosis, which is often observed in such patients. Different studies have shown that oral and gut microbiome play important roles in the development of many liver diseases. Dysbiosis has been found in the oral and gut microbiome of patients with chronic hepatitis B [52], liver cirrhosis [53], primary sclerosing cholangitis [54] and hepatocellular carcinoma (HCC) [55]. A recent epidemiological study reported a significantly increased diversity also in AIHp oral microbiome [27], with seven genera, mainly Fusobacterium, Actinomyces and Capnocytophaga, decreased in saliva of AIHp with respect to controls, and 51 genera, mainly represented by Streptococcus, Veillonella and Leptotrichia, increased. These findings suggest that elevated level of histatins in saliva of AIHp may represent a physiological protective mechanism against pathogens in the oral cavity to counter these persistent infections and inflammatory conditions. The Kendall correlation analysis performed in this study evidenced that, in AIHp group, the abundance of histatins was interrelated with that of aPRPs and statherins, which indeed have the same secretory origin and share the same physiological role regarding oral homeostasis, bacterial colonization and antimicrobial defense of the oral cavity [56].

### 3.2. Potential Salivary Biomarkers in PBCp

The panel of variated proteins and peptides in PBC, identified by Mann—Whitney tests, was constituted by S100A12, S100A9 short, cystatin S1, S2, SN and C with respect to HCs and/or AIHp. In this case, the most compact group pertains to the cystatins and S100A protein families.

Cystatins are cysteine proteases inhibitors belonging to a large super-family which comprises type 1 including cystatin A and B predominantly intracellular, type 2 including cystatins C, D, E/M, F, G and salivary S-type S, SN and SA that are all extracellular proteins, and type 3 mainly represented by kininogen. 

As cysteine proteases inhibitors, cystatins are involved in multiple processes; indeed, their down-regulation or up-regulation have been described in numerous diseases since many normal and pathological processes are coordinated by the balance between lysosomal cysteine proteases and their endogenous inhibitors cystatins. Among cystatins, our study on saliva of PBCp showed increased level of salivary S-type cystatins and cystatin C, which exert multifaceted biological functions. 

Cystatin C is a potent, reversible inhibitor in vitro of the human lysosomal cysteine proteases, mainly represented by Cathepsin-S (CTSS). CTSS plays a significant role in various intracellular and extracellular processes, including proteolysis [57] and major histocompatibility complex (MHC) class II antigen presentation, where it is important in the degradation of the invariant chain [58]. This latter aspect triggered research toward CTSS as a target in immunological disorders since inhibition of CTSS can prevent or retard MHC class II presentation and thus inflammation. As a result, inhibition of CTSS proved to have an anti-inflammatory therapeutic effect toward lupus progression [59], rheumatoid arthritis, Sjögren’s syndrome [60], encephalomyelitis [55,61] and autoimmune-triggered inflammatory responses in macrophages [62]. In the contest of liver disease, CTSS has been related to the regulation of different aspects of natural killer T (NKT) cell activation, and their inhibition prevent hepatic NKT cell expansion after lipopolysaccharides (LPS)-induced inflammation [63,64].

These evidences suggest that the increased level of cystatin C determined in PBCp group may represent a physiological protective mechanism against exacerbated expression of CTSS following immunological disorders, even if the heterogeneity of patients in terms of clinical parameters, grade of liver inflammation and stage of liver fibrosis and therapies, together with the transversal nature of the study, did not allow to demonstrate the causal link of these proteins to PBC.

In the same manner, S-type cystatins (S, SN and SA), which are of secretory origin, showed increased levels in saliva from PBCp. Cystatin S is not an inhibitor of the cysteine proteases but it is able to bind calcium, suggesting that its primary role in the oral environment is likely related to the mineral balance of the tooth [65] while cystatins SN and SA inhibit the human lysosomal cathepsins B, H and L. Interestingly, cathepsin B plays important roles in various models of liver injury, including tumor necrosis factor-α-mediated hepatocyte apoptosis [66], free fatty acid–induced liver damage [67], hepatic ischemia–reperfusion injury [68], and cholestasis [69]. Cathepsin L also plays a critical role in the pathogenesis of hepatic fibrosis since it can proteolytically degrade components of the extracellular matrix (ECM) whose accumulation in liver tissue alters the hepatic structure leading to cirrhosis [63,70]. As a result, inhibition of cathepsins B and L may be therapeutic in liver diseases. Our results evidenced an increased level of S100A12 protein in PCBp with respect to AIHp. S100A12 is a potent chemoattractant for monocytic cells [71], and it activates various cell types inducing expression of adhesion molecules and pro-inflammatory cytokines [72]. A specific involvement of this protein toward inflammatory processes of PBC has been already demonstrated since it participates in the damage of biliary epithelial cells and hepatocytes [73]. In this context, it is surprising to have found reduced levels of S100A9 proteins in saliva of PBCp because they function as cytokines and bind with Receptor for Advanced Glycation End-Products (RAGE) and Toll-like receptor-4 to activate the pro-inflammatory signaling cascade, thus increasing immune cell recruitment for their proliferation and differentiation. In this regard, it is important to highlight that almost all PBC patients recruited for this study were under UDCA therapy. UDCA is a secondary bile acid generated by the metabolism of primary bile acid, chenodeoxycholic acid, and exhibits hydrophilic and potentially cytoprotective properties. In many animal studies, UDCA induced immune suppression, cellular protection, and suppressed inflammation. It has been also demonstrated that UDCA markedly reduces ER stress, RAGE expression, and pro-inflammatory responses, including reactive oxygen species production induced in endothelial cells [74]. Correlation analysis evidenced that the cluster of type 1 cystatins and S100A9 proteins form a more compact group in the PBCp group with respect to both HCs and AIHp. These two protein families share the same non-glandular origin and resulted important for the classification of the groups based on the Boruta scores, especially for the classification of subjects in the PCBp group and in the AIHp group.

### 3.3. Classification AIH and PBC Subjects from HCs

RF analysis, one of several up-to-date machine learning methods, evidenced a panel of proteins/peptides present in saliva able to correctly classify PBCp group with respect to HCs with an 83.3% accuracy, AIHp group with respect to HCs with an 79.9% accuracy and PBCp with respect to AIHp with an 80.2% accuracy. The same statistical approach has been successfully used to classify groups of subjects based on age and health status highlighting the feasibility of the salivary proteome to discriminate groups of subjects based on physiological or pathological condition not only confined to oral cavity [75]. RF results were confirmed by LDA which provided a very good classification of subject based on PBC or AIH occurrence and with respect to HCs. Interestingly, the same protein families selected by Boruta algorithm for RF classification were also individuated as the most discriminating for the three groups in LDA. In this regard, it is important outline that strongly correlated components, which could be potentially discriminating among groups, were excluded for LDA; therefore, while it was possible to obtain an accurate classification of subjects, it was not possible to obtain an exhaustive evaluation of the discriminant power for all the components.

RF classification, which is one of the most widely used supervised machine learning algorithms for mass spectrometry data [76], was preferred over other methods because of several reasons: (a) it is not conditioned by the data distribution; (b) it has a low risk of overfitting; (c) it does not require supplementary samples for the validation of results, as each tree is built up omitting nearly one-third of the samples that are subsequently used to test the misclassification rate; (d) it provides a measure of the relative importance of each feature in the classification of samples (MDG); (e) it provides an estimation of the proximity (i.e., similarity) between each two samples that opens the possibility of applying MDS and hierarchical cluster analyses to obtain a visual representation of the classification. In addition, the use of this approach provides the possibility to correlate the position of misclassified subjects with their specific clinical profiles (i.e., severity of the disease, therapy, presence of comorbidities, etc.). These aspects will be the subject of future investigations. Surprisingly, some proteins identified as important for RF classification were not found to be significantly changed by Mann—Whitney tests. As already mentioned in the results, this apparent paradox is due to the modus operandi of RF, and more generally, of methods based on decision trees that can break variables through multiple split points to discriminate groups that would otherwise be confused, based on their averages or medians. Because of this fact, although the classification produced by RF is of considerable interest in several respects, the proteins that collectively contribute to the classification of AIHp and PBCp cannot be tout-court considered as candidate markers of the diseases. In this regard, the panel of differentially expressed proteins, identified by Mann—Whitney tests, appears to provide more reliable indications on potential biomarkers of the diseases.

### 3.4. Functional Characterization of Proteins Most Discriminating AIHp from PBCp

GO analysis was performed to increase the functional characterization of proteins mostly discriminating AIHp from PBCp groups. In AIHp only, an enrichment of terms mainly related to antimicrobial response mediated by antimicrobial peptides and antimicrobial humoral response was found. α-defensin-4 also contributes to these biological processes, being among the most discriminating proteins between AIHp and HCs in the RF analysis, even if not statistically varied according to the Mann—Whitney test results. It is noteworthy that statherin and histatins, which were identified as potential biomarkers of AIH by Mann—Whitney and Kruskal—Wallis tests, have the same biological function as antimicrobial peptides. In fact, antibacterial activities against oral pathogens have been demonstrated for statherin and its C-terminal fragments [77], which retain the specific binding sites for *Porphyromonas gingivalis*, the keystone pathogen in periodontitis [78].

GO analysis performed on proteins most discriminating PBCp with respect to HCs highlights overrepresentation of terms mainly involved in the regulation of proteolysis and negative regulation of peptidase and/or proteases. These results are also in strong agreement with those obtained with Mann—Whitney test since the proteins identified as potential biomarkers of PBC pertained to cyststin’s family, whose biological function is to inhibit lysosomal cathepsins.

### 3.5. Study Limitation

The present study represents a cross-sectional preliminary analysis of the salivary proteome in patients affected by autoimmune liver diseases; therefore, the low number of subjects involved in the study may not be entirely representative of the considered population. Nevertheless, the statistical analysis performed by our group provides a good classification of subjects based on AIH or PBC occurrence. A larger population will be useful to further validate our findings. We must also acknowledge that most of our ALD patients were undergoing therapy at the time of saliva and blood sample collection; therefore, we could not assess the potential impact of such treatments on the salivary proteome since we have limited data regarding untreated patients. Future prospective studies enrolling a higher number of patients at the time of ALD diagnosis with different grade of liver inflammation and stage of liver fibrosis could help clarify whether salivary proteome might correlate with liver disease severity and ultimately response to therapy. Moreover, the transversal nature of the study did not allow for a demonstration of the causal link of the selected proteins to ALD, which can be considered potential biomarkers rather than causal mediators of these pathologies.

## 4. Materials and Methods

### 4.1. Ethical Statement

This is a cross-sectional study performed in 2021 on AIHp and PBCp recruited from the liver unit of University Hospital of Cagliari, Sardinia, Italy. Patients and healthy controls signed the informed written consent that agreed with the latest stipulations established by the Declaration of Helsinki. The Committee of the “Azienda Ospedaliero-Universitaria di Cagliari”, Cagliari, Italy, approved the study on 21 July 2021 (reference number PG/2021/11303). 

### 4.2. Study Subjects and Clinical Studies

Patients were diagnosed based on the criteria reviewed by the International Autoimmune Hepatitis Study Group (IAIHG) in 1999 [79] and by the EASL clinical practice guidelines [7]. Were included in the study patients showing, at the time of saliva sampling almost normal values of ALT and AST;; based on this inclusion criteria, only patients that were under pharmacological therapy for at least three years were selected. Only two AIHp were without therapy but were included in this study because of their normal values of transaminases. Patients affected by Overlap syndrome, chronic hepatitis induced by HBV or HCV, drug or alcohol abuse, fatty liver disease, primary sclerosing cholangitis and any major oral disease (periodontitis, caries) were excluded. 

The control group (HCs) included age and sex matched healthy volunteers recruited from the local population. Controls were excluded if they were relatives of the patients and had a history of liver diseases, immunological disorders and major oral diseases. Most of the controls were patient’s caregivers recruited in the hospital during follow-up and/or medical and research personnel involved in the study.

### 4.3. Sample Collection and Treatment

Unstimulated whole saliva (WS) (from 0.2 to 1 mL) was collected with a soft plastic aspirator at the basis of the tongue from 9 to 13 a.m. in fasting conditions using a standard protocol optimized to preserve salivary proteins from proteolytic degradation. After collection, samples were immediately mixed with an equal volume of 0.2% (*v/v*) 2,2,2-trifluoroacetic acid (TFA) containing 50 μM of leu-enkephalin as internal standard and centrifuged at 14,000 RPM for 10 min at 4 °C. The acidic-soluble fraction of whole saliva (supernatant) was collected and stored at −80 °C until the analysis. The total protein concentration (TPC) of each sample was determined in duplicate by the bicinchoninic acid (BCA) Protein Assay kit (Pierce™ BCA Protein Assay Kit, Thermo Fisher Scientific, Waltham, MA, USA), following the provided instructions.

### 4.4. RP-HPLC ESI-MS Analysis

All the chemicals and reagents used for analysis were purchased from Sigma Aldrich (St. Louis, MO, USA). Peptides and proteins search and label-free quantification was performed by RP-HPLC low-resolution ESI-MS. 35 μL of the acidic-soluble fraction of WS from each sample were analyzed by a Surveyor HPLC system connected to a LCQ advantage mass spectrometer (Thermo Fisher Scientific, CA, USA) equipped with an ESI source. The chromatographic column was a Vydac (Hesperia, CA, USA) C8 column with 5 μm particle diameter (150 × 2.1 mm). The following solutions were utilized for the separation: (eluent A) 0.056% (*v/v*) aqueous TFA and (eluent B) 0.05% (*v/v*) TFA in acetonitrile/water 80/20. A linear gradient was applied from 0 to 55% of B in 40 min, and from 55% to 100% of B in 10 min, at a flow rate of 0.10 mL/min toward the ESI source. During the first 5 min of separation, the eluate was diverted to waste to avoid instrument damage because of the high salt concentration. Mass spectra were collected every 3 ms in the m/z range 300–2000 in positive ion mode. The MS spray voltage was set to 5.0 kV and the capillary temperature to 260 °C. MS resolution was 6000. Deconvolution of averaged ESI-MS spectra was performed by MagTran 1.0 software [80]. 

### 4.5. Data Analysis and Quantification

Experimental average mass values (Mav) of salivary proteins and peptides characterized in previous studies [41,42] were compared with theoretical average mass values (Mav) available at Swiss-Prot Data Bank (http://www.uniprot.org/, accessed on 1 May 2022). The label-free quantitation of peptides and proteins was performed by measuring the area of RP-HPLC low-resolution ESI-MS eXtracted Ion Current (XIC) peaks generated by selecting specific *m/z* ions for each protein/peptide, considered when the S/N ratio was at least 5. Peak characteristics should satisfy the following parameters: baseline window 15, area noise factor 50, peak noise factor 50, peak height 15% and tailing factor 1.5. Area of the XIC peaks, expressed by arbitrary units, is proportional to the protein concentration, and, under constant analytical conditions, it allows performing relative quantification of the same protein in different samples and quantifies an indefinite number of proteins/peptides in a unique analysis [81,82]. The estimated percentage error of the XIC analysis was < 8%. 

Eventual dilution errors occurring during sample collection were adjusted by correcting XIC peak areas of each peptide/protein with the XIC peak area of the leu-enkephalin 50 µM used as internal standard in the aqueous 0.2% trifluoroacetic solution added to whole saliva in ratio 1:1 (*v/v*) at the collection time [42]. The following correction equation was applied: corrected area of protein = measured area of protein ∗ (expected area of leu-enkephalin 50 mM/measured area of leu-enkephalin). In addition, the TPC in g/L of every single sample was used to normalize the XIC peak areas of each peptide/protein detected as follows: the value of the XIC peak area corrected with leu-enkephalin was divided by the TPC [40].

### 4.6. Statistical Analysis

According to Kolmogorov—Smirnov and other goodness-of-fit tests (Shapiro—Wilk, Anderson—Darling, Lilliefors, with *p*-values < 0.0001 in almost all tests), distribution of XIC peak areas of all proteins/peptides showed a considerable deviation from normality. Thus, the non-parametric Mann—Whitney test, Kruskal—Wallis test, and Kendall correlation were adopted. 

The difference of TPC within the three groups was tested by both Mann—Whitney and Kruskal—Wallis tests followed by Dunn’s post-hoc tests using GraphPad Prism software (version 5.0). 

Statistical analysis considered both single proteoforms and the sum of different proteoforms of the same protein; for simplicity, both single and summed proteoforms are named “components” in the text. The number of components examined in this study is 71. MS data were analyzed using different statistical methods: (a) Mann—Whitney and Kruskal—Wallis tests, (b) Kendall correlation, (3) random forest (RF) and (4) linear discriminant analysis (LDA). Significant *p*-values of multiple Mann—Whitney tests were verified by the Benjamini—Hochberg procedure [83] to maintain a false discovery rate of 0.05. Multidimensional scaling (MDS) was applied to Kendall correlations to obtain a dimensionally reduced diagram of co-expressed proteins. For RF analysis, algorithm parameters, such as the number of trees to grow and the number of features randomly sampled for each split, were preliminarily tuned to minimize the classification error. RF was applied to three mixed data sets: (1) HCs and AIHp; (2) HCs and PBCp; (3) AIHp and PBCp. Classification accuracy was calculated as the proportion of correct assessments (both true positive and true negative) to the total number of assessments. The Boruta algorithm [84] was used to select a subset of components to increase classification accuracy. Diagrams of classified subjects were obtained by MDS using the RF proximity values (the proximity between two subjects is the normalized frequency of trees that contain the two subjects in the same end node). LDA was applied to the mixed data set formed by the three groups taken together: HCs, PBCp and AIHp. 

Multivariate analyses were made using R (RCoreTeam. R: A language and environment for statistical computing. Vienna, Austria: R Foundation for Statistical Computing; 2014. http://www.R-project.org/ (accessed on 1 August 2022)) and (XLSTAT 2007; Statistical Software for Excel. https://www.xlstat.com (accessed on 1 August 2022).

### 4.7. Gene Ontology Enrichment Analyses

To increase the functional characterization of varied proteins between groups, a gene ontology (GO) Enrichment analysis based on biological process (BP) was performed [85,86,87]. The most discriminant components between groups resulting by Boruta algorithm, and used for RF classification, were submitted to a Fisher’s exact test with FDR correction versus *homo sapiens* reference list. The statistical analysis was performed with the tool PANTHER Overrepresentation Test (version 17.0, Released 12 July 2022) and GO Ontology database (Released 1 July 2022; DOI: 10.5281/zenodo.6799722, http://geneontology.org/, last accessed on 7 October 2022). Statistical significance was set to 0.05 (*p*-value ≤ 0.05). 

## 5. Conclusions

The top-down proteomic pipeline exploited in this study allowed evidencing, for the first time, the qualitative and quantitative differences of targeted salivary proteins/peptides in AIH and PCB patients with respect to healthy controls.

Despite the high heterogeneity of the clinical manifestation of the patients, the robustness of the analytical and statistical approach used allowed highlighting a set of potential salivary biomarkers of AIH and PBC. Biomarker candidates of AIHp were individuated in peptides/proteins involved in antimicrobial defense, while peptides/proteins involved in innate immune system characterized PBCp.

RF analysis revealed the feasibility of the salivary proteome to discriminate groups of subjects based on AIH or PBC occurrence. In this regard, RF, strengthened by LDA, appears an attractive machine-learning tool suited for classification of AIH and PBC based on their different salivary proteomic profile.

In our opinion, the results suggest that differences found in the salivary proteomic profile of AIH and PBC may reflect the immuno-pathological differences between the two diseases, even with respect to controls, rather than the clinical course of the disease and/or iatrogenic factors.

## Figures and Tables

**Figure 1 ijms-24-00959-f001:**
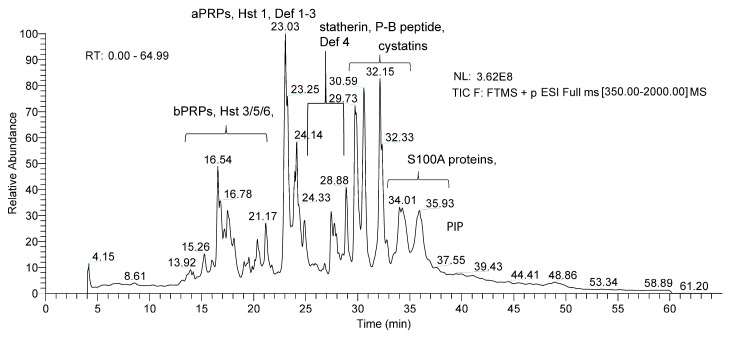
Typical total ion current (TIC) chromatographic profile of acidic-soluble fraction of adult saliva obtained by RP-HPLC-ESI-low-resolution MS analysis and indication of names and elution ranges of the protein families investigated.

**Figure 2 ijms-24-00959-f002:**
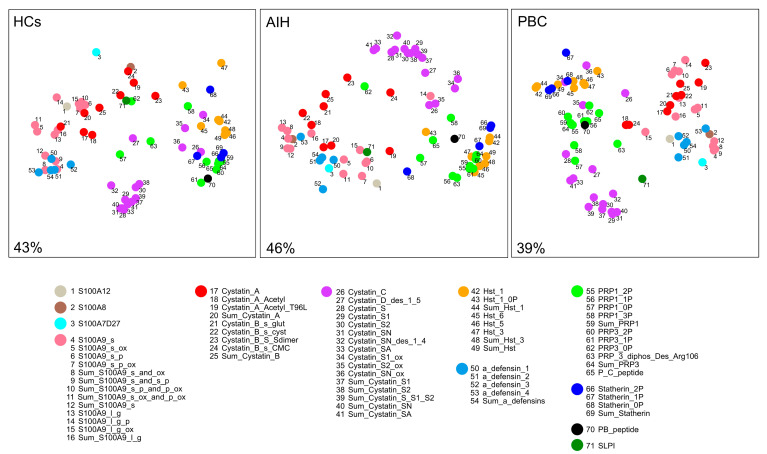
Multidimensional scaling (MDS) diagrams of Kendall correlations among protein levels in the HCs, AIHp and PBCp groups. To facilitate the understanding of the diagrams, the 71 components were grouped into different categories, numerically and color encoded, based on their structural/functional similitudes and secretory origin. The degree of clustering of points accounts for the degree of proteins/peptides co-expression. Percent values indicate how much two-dimensional diagrams represent the information contained in the whole multi-dimensional structure.

**Figure 3 ijms-24-00959-f003:**
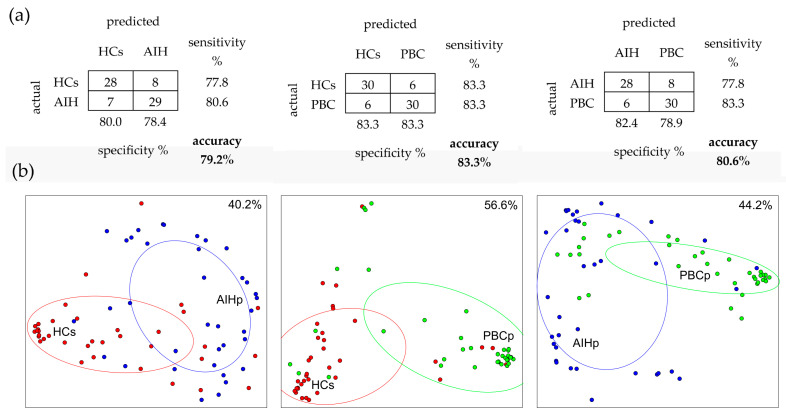
RF applied to three mixed data sets. (**a**) Confusion matrices of RF classifications validated by out-of-bag samples. Matrix rows represent the actual classes, while columns represent the predicted classes. Marginal rows show the frequency of false positives, while marginal columns show the frequency of false negatives. The overall accuracy considers both true positives and true negatives. (**b**) Multidimensional Scaling (MDS) diagrams showing the relationship among subjects, using the proximity values calculated by RF. Each group is delimited by a dispersion ellipse with a confidence of 1.6 standard deviations. Percent values indicate how much two-dimensional diagrams represent the information contained in the whole multi-dimensional structure.

**Figure 4 ijms-24-00959-f004:**
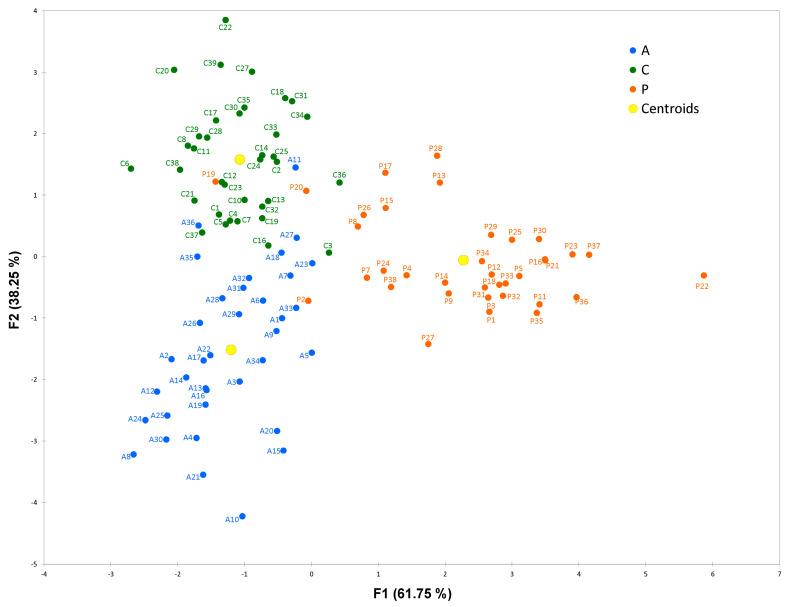
LDA classification of HCs, AIHp and PBCp subjects. Centroids represent the average subject position of each group.

**Table 1 ijms-24-00959-t001:** Clinical features and pharmacological treatment of AIHp and PBCp included in the study measured at the time of saliva sampling.

Parameters		AIHp	PBCp
Age, Average (range)	Years	54.53 (29.81–74.89)	65.13 (41.27–81.15)
Gender, *n* (%)	Female	32 (88.8%)	35 (97.2%)
BMI, Average (range)	Kg/m^2^	25.68 (17.57–38.45)	24.72 (19.10–40.43)
Cirrhosis, *n* (%)		7 (19.4%)	4 (11.1%)
Histological stage, *n* (%)	I	9 (25%)	15 (41.6%)
	II	4 (11.1%)	8 (22.2%)
	III	6 (16.6%)	1 (2.7%)
	IV	4 (11.1%)	4 (11.1%)
	Not available	11 (30.5%)	10 (27.7%)
Positivity to autoantibodies, *n* (%)	ANA	25 (69.4%)	30 (83.3%)
	SMA	20 (55.5%)	7 (19.4%)
	LKM	3 (8.3%)	1 (2.7%)
AST, Median (range)	IU/L	23.5 (13–57)	27.0 (16–71)
ALT, Median (range)	IU/L	21.0 (5–58)	23.0 (11–78)
GGT, Median (range)	IU/L	25.5 (6–167)	42.0 (12–167)
ALP, Median (range)	IU/L	67.0 (28–216)	107.0 (52–222)
IgG, Median (range)	g/dL	1.4 (0.69–2.51)	1.4 (0.7–2.3)
Albumin, Median (range)	g/dL	3.9 (1.2–4.83)	3.9 (2.8–4.3)
Prothrombin time, Median (range)	INR	0.97 (0.92–1.06)	1.01(0.86–1.81)
TB, Median (range)	mg/dL	0.7 (0.25–2.19)	0.6 (0.34–2.95)
Platelets, Median (range)	109/L	217.5 (91–423)	242 (46–418)
Pharmacological treatment (% treated)	Azathioprine + Steroids	41%	n.a.
	Steroids	25.5%	n.a.
	Azathioprine	17.6%	n.a.
	Naïve	5.5%	n.a.
	UDCA	n.a.	100%

n.a. not applicable; BMI: body mass index; ANA: antinuclear antibodies, SMA: smooth muscle antibodies; LKM: renal microsomal antigen antibodies; AST: aspartate aminotransferase; ALT: alanine aminotransferase; GGT: γ-glutamyl transferase; ALP: alkaline phosphatase, IgG: immunoglobulin G; TB: total bilirubin; UDCA: Ursodeoxycholic Acid.

**Table 2 ijms-24-00959-t002:** Comparisons between HCs, AIHp and PBCp. For each pairwise comparison, two columns are reported: the first column shows the Mann—Whitney significant *p*-values, highlighted by color tones ranging from yellow to red; the second column shows the direction of the change. The last column on the right shows the results of the ANOVA-analogue nonparametric Kruskal—Wallis test.

Components		HCs vs. AIHp		HCs vs. PBCp		AIHp vs. PBCp		AIHp vs. PBCp vs. HCs
N	Description	Mann—Whitney		Mann—Whitney		Mann—Whitney		Kruskal—Wallis
		*p*-Value	Change	*p*-Value	Change	*p*-Value	Change	*p*-Value
1	S100A12					<0.05	PBC > AIH	<0.05
2	S100A8							
3	S100A7D27					<0.05	PBC > AIH	<0.01
4	S100A9_short							
5	S100A9_short_ox			<0.05	C > PBC			<0.05
6	S100A9_short_P							<0.05
7	S100A9_short_P_ox							
8	Sum_S100A9_short_and_ox							
9	Sum_S100A9_s_and_s_P							
10	Sum_S100A9_s_P_and_P_ox							
11	Sum_S100A9_s_ox_and_P_ox			<0.05	C > PBC			<0.05
12	Sum_S100A9_short			<0.05	C > PBC			
13	S100A9_long_g							
14	S100A9_long_g_p							
15	S100A9_long_g_ox							<0.05
16	Sum_S100A9_long_g							
17	Cystatin_A	<0.05	AIH > C			<0.05	AIH > PBC	<0.05
18	Cystatin_A_Acetyl							
19	Cystatin_A_Acetyl_T96L							
20	Sum_Cystatin_A	<0.05	AIH > C					
21	Cystatin_B_S_glut							
22	Cystatin_B_S_cyst							
23	Cystatin_B_SSdimer							
24	Cystatin_B_S_CMC							
25	Sum_Cystatin_B							
26	Cystatin_C			<0.05	PBC > C			<0.01
27	Cystatin_D_des_1_5							
28	Cystatin_S							
29	Cystatin_S1			<0.001	PBC > C	<0.01	PBC > AIH	<0.001
30	Cystatin_S2			<0.0001	PBC > C	<0.01	PBC > AIH	<0.0001
31	Cystatin_SN			<0.01	PBC > C	<0.05	PBC > AIH	<0.05
32	Cystatin_SN_des_1_4							
33	Cystatin_SA							
34	Cystatin_S1_ox	<0.001	AIH > C					<0.01
35	Cystatin_S2_ox							
36	Cystatin_SN_ox							
37	Sum_Cystatin_S1			<0.0001	PBC > C	<0.01	PBC > AIH	<0.0001
38	Sum_Cystatin_S2			<0.0001	PBC > C	<0.01	PBC > AIH	<0.0001
39	Sum_Cystatin_S_S1_S2			<0.0001	PBC > C	<0.001	PBC > AIH	<0.0001
40	Sum_Cystatin_SN			<0.01	PBC > C			<0.05
41	Sum_Cystatin_SA							
42	Hst_1							
43	Hst_1_0P							
44	Sum_Hst_1							
45	Hst_6	<0.01	AIH > C			<0.05	AIH > PBC	
46	Hst_5	<0.05	AIH > C			<0.05	AIH > PBC	<0.05
47	Hst_3	<0.05	AIH > C			<0.01	AIH > PBC	<0.01
48	Sum_Hst_3	<0.01	AIH > C			<0.05	AIH > PBC	<0.05
49	Sum_Hst	<0.05	AIH > C					<0.05
50	α_defensin_1							
51	α_defensin_2							
52	α_defensin_3							
53	α_defensin_4							
54	Sum_α_defensins							
55	PRP1_2P					<0.05	AIH > PBC	
56	PRP1_1P							
57	PRP1_0P							
58	PRP1_3P			<0.01	PBC > C			<0.05
59	Sum_PRP1					<0.05	AIH > PBC	
60	PRP3_2P							
61	PRP3_1P			<0.05	C > PBC	<0.05	AIH > PBC	<0.05
62	PRP3_0P							
63	PRP_3_diphos_Des_Arg106					<0.05	PBC > AIH	<0.05
64	Sum_PRP3							
65	P_C_peptide							
66	Statherin_2P	<0.05	AIH > C					
67	Statherin_1P	<0.05	AIH > C			<0.05	AIH > PBC	
68	Statherin_0P							
69	Sum_Statherin	<0.05	AIH > C					
70	PB_peptide					<0.01	AIH > PBC	<0.05
71	SLPI			<0.05	PBC > C			<0.05

**Table 3 ijms-24-00959-t003:** GO biological process enrichment of the most discriminant components of HCs-AIHp and HCs-PBCp mixed data sets results by Boruta algorithm. For each group, only the first ten biological processes are shown with the relative *p*-values and the number of associated proteins in respect to the total number of components (nine for HCs-AIHp and six for HCs-PBCp) used for analysis.

**HCs-AIHp Mixed Data Set**		
**GO Biological Process**	**No. Associated Proteins**	**Enrichment *p*-Value**
defense response (GO:0006952)	6/9	9.62 × 10^−6^
antimicrobial humoral immune response mediated by antimicrobial peptide (GO:0061844)	5/9	4.69 × 10^−10^
antimicrobial humoral response (GO:0019730)	5/9	2.28 × 10^−9^
humoral immune response (GO:0006959)	5/9	1.51 × 10^−7^
defense response to bacterium (GO:0042742)	5/9	2.50 × 10^−7^
regulation of endopeptidase activity (GO:0052548)	5/9	4.40 × 10^−7^
regulation of peptidase activity (GO:0052547)	5/9	6.16 × 10^−7^
response to bacterium (GO:0009617)	5/9	6.95 × 10^−6^
regulation of proteolysis (GO:0030162)	5/9	7.47 × 10^−6^
regulation of hydrolase activity (GO:0051336)	5/9	3.22 × 10^−5^
**HCs-PBCp Mixed Data Set**		
regulation of peptidase activity (GO:0052547)	5/6	3.10 × 10^−8^
regulation of endopeptidase activity (GO:0052548)	5/6	2.21 × 10^−8^
regulation of proteolysis (GO:0030162)	5/6	3.63 × 10^−7^
regulation of hydrolase activity (GO:0051336)	5/6	1.74 × 10^−6^
negative regulation of peptidase activity (GO:0010466)	4/6	3.71 × 10^−7^
negative regulation of endopeptidase activity (GO:0010951)	4/6	3.22 × 10^−7^
regulation of cysteine-type endopeptidase activity (GO:2000116)	4/6	2.78 × 10^−7^
negative regulation of proteolysis (GO:0045861)	4/6	1.18 × 10^−6^
negative regulation of hydrolase activity (GO:0051346)	4/6	1.45 × 10^−6^
negative regulation of catalytic activity (GO:0043086)	4/6	2.96 × 10^−5^

## Data Availability

The data presented in this study are available in Appendix A.

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
