# Peer review of "Top-Down Proteomics Detection of Potential Salivary Biomarkers for Autoimmune Liver Diseases Classification"

_ijms, 2023, doi:10.3390/ijms24020959_

Round 1
Reviewer 1 Report
INTRODUCTION
- Overall, the authors clearly present the main concepts related to AIH, PBC, and PSC. St the beginning of the introduction, the authors correctly highlight that “Autoimmune liver diseases (AILDs) include a series of pathological conditions that target the liver and have a wide spectrum of presentation, ranging from asymptomatic forms to end stage liver disease requiring liver transplantation”. I think the authors (perhaps, after line 65) could also mention all those conditions and/or comorbidity which can be associated to an immune-mediated liver/biliary/cholecystic pathology, which can additionally complicate the differential diagnosis, such as celiac disease, for instance (see: Dig Liver Dis. 2016 Feb;48(2):112-9. doi: 10.1016/j.dld.2015.11.013; Nutrients. 2022 Oct 19;14(20):4379. doi: 10.3390/nu14204379.
- The general objectives are clearly stated at the end of the introduction.
RESULTS
- I think that the first part of the results should be also include in a specific subsection with its own subheading.
- The authors should report the main demographic, clinical and laboratory characteristics mentioned in section 2.1 in an additional and specific table, which could show all the study population and the different groups.
- Except for these aspects, the results section is well-organized and complete.
METHODS
- section 4.1: As mentioned above, the demographics should be included in the results, along with the description of clinical (also therapeutic) and laboratory aspects.
- I would also suggest creating a specific subsection for the ethical statement. In addition to the IRB approval, I recommend including also a statement related to patients’ informed consent.
- Moreover, the authors should clearly define the healthy volunteers/controls: it is not clear who they are, how/where exactly they were recruited, etc.
DISCUSSION
- There are many aspects deserving discussion. In the current version, this sections sounds dispersive. Therefore, I suggest the authors start the discussion by schematically listing and highlighting their main findings. Moreover, I would recommend to re-organize the discussion in subsections as well or at least clearly highlight the specific discussion of each point that should be listed at the beginning of the introduction.
- there is no discussion regarding the study limitations. A specific paragraph should be added.
Author Response
Overall, the authors clearly present the main concepts related to AIH, PBC, and PSC. St the beginning of the introduction, the authors correctly highlight that “Autoimmune liver diseases (AILDs) include a series of pathological conditions that target the liver and have a wide spectrum of presentation, ranging from asymptomatic forms to end stage liver disease requiring liver transplantation”. I think the authors (perhaps, after line 65) could also mention all those conditions and/or comorbidity which can be associated to an immune-mediated liver/biliary/cholecystic pathology, which can additionally complicate the differential diagnosis, such as celiac disease, for instance (see: Dig Liver Dis. 2016 Feb;48(2):112-9. doi: 10.1016/j.dld.2015.11.013; Nutrients. 2022 Oct 19;14(20):4379. doi: 10.3390/nu14204379.
Reply: as suggested by the referee we added in the introduction section a paragraph explaining the most common comorbidities linked to AIH and PBC. In particular, we insert a comment and a new reference focused on celiac disease.
- The general objectives are clearly stated at the end of the introduction.
RESULTS
- I think that the first part of the results should be also include in a specific subsection with its own subheading.
Reply: the first part of the results has been included into a specific subsection titled: “2.1. Top-Down Mass Spectrometry pipeline”. Consequently, all the other subsections have been renumbered.
- The authors should report the main demographic, clinical and laboratory characteristics mentioned in section 2.1 in an additional and specific table, which could show all the study population and the different groups.
Reply: the demographic data for each HCs, AIHp and PBCp involved in the study are reported in table S2 while the clinical features and pharmacological treatment of AIHp and PBCp included in the study were reported in Table S3 but, following the referee’s indication, it has moved from supplemental materials to results and numbered as Table 1. However, for completeness the main demographic features of subjects included in the study has been also reported at the beginning of section 2.2 (previously 2.1). The new table 1 contains more information with respect to the previous version (e.i. Cirrhosis, Histological stage and positivity of autoantibodies of PBC patients). Moreover, we also correct a mistake regarding the number of AIH patients without pharmacological treatment (5.5% instead of 17%)
- Except for these aspects, the results section is well-organized and complete.
 
METHODS
- section 4.1: As mentioned above, the demographics should be included in the results, along with the description of clinical (also therapeutic) and laboratory aspects.
Reply: done
- I would also suggest creating a specific subsection for the ethical statement. In addition to the IRB approval, I recommend including also a statement related to patients’ informed consent.
Reply: As requested a new paragraph, reporting the ethical statement, has been added and numbered as 4.1, consequently all the other paragraphs have been renumbered.
- Moreover, the authors should clearly define the healthy volunteers/controls: it is not clear who they are, how/where exactly they were recruited, etc.
Reply: All the controls were age and sex matched volunteers recruited from the local healthy population. The demographic features of all controls are reported in table S2. A short sentence on the control group has been reported in paragraph 4.2.
DISCUSSION
- There are many aspects deserving discussion. In the current version, this section sounds dispersive. Therefore, I suggest the authors start the discussion by schematically listing and highlighting their main findings. Moreover, I would recommend to re-organize the discussion in subsections as well or at least clearly highlight the specific discussion of each point that should be listed at the beginning of the introduction.
Reply: As suggested by this referee the discussion has been deeply reorganized. The first part schematically introduces the different tasks that will be further discussed. Following this scheme, the discussion has been divided into different subsections where each point introduced in the first part is discussed in detail.
- there is no discussion regarding the study limitations. A specific paragraph should be added.
Reply: according to the referee’s suggestion a new subsection titled “ study limitation and future perspectives” has been added a the end of the discussion and numbered as 3.5.
Reviewer 2 Report
The idea of this study to evidence, by a top-down proteomic pipeline possible qualitative and/ or quantitative differences of targeted salivary proteins/peptides, in patients with either autoimmune hepatitis or primary biliary cholangitis compared with healthy controls is very interesting and with important clinical implications. Even though this study included a relatively small number of patients, I consider that the findings are interesting and that the results obtained can significantly contribute to further large studies. However, I recommend emphasizing much better the novelty, the original elements, and also the limitations of the study.
Author Response
The idea of this study to evidence, by a top-down proteomic pipeline possible qualitative and/ or quantitative differences of targeted salivary proteins/peptides, in patients with either autoimmune hepatitis or primary biliary cholangitis compared with healthy controls is very interesting and with important clinical implications. Even though this study included a relatively small number of patients, I consider that the findings are interesting and that the results obtained can significantly contribute to further large studies. However, I recommend emphasizing much better the novelty, the original elements, and also the limitations of the study.
Reply: We thank the referee for its encouraging comments. The first part of the discussion has been improved highlighting the novelty of our study. A new paragraph reporting the study’s limitations and future perspectives has been added at the end of the discussion (subsection 3.5).
Reviewer 3 Report
In this study the authors tried to assess the possible use of saliva as a diagnostic tool in autoimmune hepatitis (AIH) and primary biliary cholangitis (PBC) being both characterized by chronic hepatic inflammation and progressive liver fibrosis. Salivary proteomic data of 36 healthy controls (HCs), 36 AIH and 36 PBC patients, obtained by liquid chromatography/mass spectrometry top-down pipeline, were analyzed by multiple Mann-Whitney test, Kendall correlation, Random-Forest (RF) analysis and linear discriminant analysis (LDA).
They found indications on the panel of differentially expressed salivary proteins and peptides, namely cystatin A, statherin, histatin 3, histatin 5 and histatin 6 which were elevated in AIH patients with respect to both HCs and PBC patients, while S100A12, S100A9 short, cystatin S1, S2, SN and C showed varied levels in PBC with respect to HCs and/or AIH patients. RF analysis evidenced a panel of salivary proteins/peptides able to classify with good accuracy PBC vs HCs (83,3%), AIH vs HCs (79,9%) and PBC vs AIH (80,2%). They concluded that RF appears an attractive machine-learning tool suited for the classification of AIH and PBC based on their different salivary proteomic profiles.
The study is of interest providing a novel approach for autoimmune liver disease profiling. However, several points deserve further details and should be addressed:
-saliva: it would significant to specify when samples were collected (at diagnosis? during the follow-up? under treatment? which treatment). Could this affect study results?
-the authors discuss in the introduction the need and relevance of peripheral biomarkers for the proper management of autoimmune liver diseases. However, they should discuss the clinical relevance of the diagnostic role of very specific serum autoantibodies. In particular, for AIH, antinuclear antibodies, in particular, those with "homogenous" immunofluorescence pattern, and smooth muscle antibodies SMA), in particular those with anti-actin specificity, have high AIH-specificity as previously demonstrated (Diagnosis and therapy of autoimmune hepatitis. Mini Rev Med Chem. 2009 Jun;9(7):847-60; Antibodies to filamentous actin (F-actin) in type 1 autoimmune hepatitis. J Clin Pathol. 2006 Mar;59(3):280-4).
-Also PBC patients may be easily identified by the detection of very disease-specific autoantibodies such as antimitochondrial and the so-called PBC-specific antinuclear antibodies (Antinuclear antibodies giving the 'multiple nuclear dots' or the 'rim-like/membranous' patterns: diagnostic accuracy for primary biliary cirrhosis. Aliment Pharmacol Ther. 2006 Dec;24(11-12):1575-83; Autoantibodies to speckled protein family in primary biliary cholangitis. Allergy Asthma Clin Immunol. 2021 Mar 31;17(1):35; ).
-the authors could suggest how salivary biomarkers could be useful in the follow-up of autoimmune liver diseases patients (correlation with treatment response?).
Author Response
In this study the authors tried to assess the possible use of saliva as a diagnostic tool in autoimmune hepatitis (AIH) and primary biliary cholangitis (PBC) being both characterized by chronic hepatic
inflammation and progressive liver fibrosis. Salivary proteomic data of 36 healthy controls (HCs), 36 AIH and 36 PBC patients, obtained by liquid chromatography/mass spectrometry top-down pipeline, were analyzed by multiple Mann-Whitney test, Kendall correlation, Random-Forest (RF) analysis and linear discriminant analysis (LDA).
They found indications on the panel of differentially expressed salivary proteins and peptides, namely cystatin A, statherin, histatin 3, histatin 5 and histatin 6 which were elevated in AIH patients with respect to both HCs and PBC patients, while S100A12, S100A9 short, cystatin S1, S2, SN and C showed varied levels in PBC with respect to HCs and/or AIH patients. RF analysis evidenced a panel of salivary proteins/peptides able to classify with good accuracy PBC vs HCs (83,3%), AIH vs HCs (79,9%) and PBC vs AIH (80,2%). They concluded that RF appears an attractive machine-learning tool suited for the classification of AIH and PBC based on their different salivary proteomic profiles.
The study is of interest providing a novel approach for autoimmune liver disease profiling. However, several points deserve further details and should be addressed:
-saliva: it would significant to specify when samples were collected (at diagnosis? during the follow-up? under treatment? which treatment). Could this affect study results?
Reply: we thank the referee for this comment because allowed us to improve this very important aspect of the work. Saliva sampling was performed during follow-up in subjects that were undergoing therapy showing, at the moment of sampling, normal values of ALT and AST. All the PBC patients were under UDCA therapy and this aspect has been also discussed in the discussion section. AIH patients, as reported in the new table 1 (previously named Table S3), were under different pharmacological therapy but they were included in this study because of their normal values of transaminases. To our knowledge there is no clinical study assessing the variability of the salivary proteome in AILDs due to pharmacological therapy. This could be an aspect deserving further investigations and it has been included in the new paragraph at the end of discussion titled “Study limitation and future perspectives”. For completeness the paragraph “Characteristics of the Participants and saliva sampling” under Results section has been implemented with these clinical aspects and Table S3, reporting the clinical features and pharmacological treatment of AIHp and PBCp enrolled in this study, has moved from supplemental materials to results and numbered as Table 1.
-the authors discuss in the introduction the need and relevance of peripheral biomarkers for the proper management of autoimmune liver diseases. However, they should discuss the clinical
relevance of the diagnostic role of very specific serum autoantibodies. In particular, for AIH, antinuclear antibodies, in particular, those with "homogenous" immunofluorescence pattern, and smooth muscle antibodies SMA), in particular those with anti-actin specificity, have high AIH-specificity as previously demonstrated (Diagnosis and therapy of autoimmune hepatitis. Mini Rev Med Chem. 2009 Jun;9(7):847-60; Antibodies to filamentous actin (F-actin) in type 1 autoimmune hepatitis. J Clin Pathol. 2006 Mar;59(3):280-4).
-Also PBC patients may be easily identified by the detection of very disease-specific autoantibodies such as antimitochondrial and the so-called PBC-specific antinuclear antibodies (Antinuclear antibodies giving the 'multiple nuclear dots' or the 'rim-like/membranous' patterns: diagnostic accuracy for primary biliary cirrhosis. Aliment Pharmacol Ther. 2006 Dec;24(11-12):1575-83; Autoantibodies to speckled protein family in primary biliary cholangitis. Allergy Asthma Clin Immunol. 2021 Mar 31;17(1):35; ).
Reply: following the referee’s suggestions we have improved the introduction by adding the main clinical relevance of the diagnostic role of very specific serum autoantibodies for both AIH and PBC.
-the authors could suggest how salivary biomarkers could be useful in the follow-up of autoimmune liver diseases patients (correlation with treatment response?). 
Reply: Most of our ALD patients were undergoing therapy at the time of saliva and blood sample collection, therefore we could not assess the potential impact of such treatments on the salivary proteome since we have limited data regarding untreated patients. Future prospective studies enrolling a higher number of patients at the time of ALD diagnosis with different grade of liver inflammation and stage of liver fibrosis could help clarify whether salivary proteome might correlate with liver disease severity and ultimately to response to therapy. These questions have been addressed in the new paragraph 3.5 titled “Study limitations and future perspectives”.
Round 2
Reviewer 1 Report
Overall, the authors clearly present the main concepts related to AIH, PBC, and PSC. St the beginning of the introduction, the authors correctly highlight that “Autoimmune liver diseases (AILDs) include a series of pathological conditions that target the liver and have a wide spectrum of presentation, ranging from asymptomatic forms to end stage liver disease requiring liver transplantation”. I think the authors (perhaps, after line 65) could also mention all those conditions and/or comorbidity which can be associated to an immune-mediated liver/biliary/cholecystic pathology, which can additionally complicate the differential diagnosis, such as celiac disease, for instance (see: Dig Liver Dis. 2016 Feb;48(2):112-9. doi: 10.1016/j.dld.2015.11.013; Nutrients. 2022 Oct 19;14(20):4379. doi: 10.3390/nu14204379.
Reply: as suggested by the referee we added in the introduction section a paragraph explaining the most common comorbidities linked to AIH and PBC. In particular, we insert a comment and a new reference focused on celiac disease.
RR- The authors added a paragraph. Revision accepted. However, the authors could highlight more the challenges for differential diagnosis in presence of comorbidities, especially if the related immunopathological processes may also affect other tract of the hepatobiliary system.
RESULTS
- I think that the first part of the results should be also include in a specific subsection with its own subheading.
Reply: the first part of the results has been included into a specific subsection titled: “2.1. Top-Down Mass Spectrometry pipeline”. Consequently, all the other subsections have been renumbered.
RR- Accepted.
- The authors should report the main demographic, clinical and laboratory characteristics mentioned in section 2.1 in an additional and specific table, which could show all the study population and the different groups.
Reply: the demographic data for each HCs, AIHp and PBCp involved in the study are reported in table S2 while the clinical features and pharmacological treatment of AIHp and PBCp included in the study were reported in Table S3 but, following the referee’s indication, it has moved from supplemental materials to results and numbered as Table 1. However, for completeness the main demographic features of subjects included in the study has been also reported at the beginning of section 2.2 (previously 2.1). The new table 1 contains more information with respect to the previous version (e.i. Cirrhosis, Histological stage and positivity of autoantibodies of PBC patients). Moreover, we also correct a mistake regarding the number of AIH patients without pharmacological treatment (5.5% instead of 17%)
RR- The authors added the requested information and tables. Accepted.
METHODS
- section 4.1: As mentioned above, the demographics should be included in the results, along with the description of clinical (also therapeutic) and laboratory aspects.
Reply: done
RR- Some demographic information is still present in this section, whereas numbers should be part of the result section only. This section should clearly and simply state the inclusion criteria used to identify the study population and, if any, exclusion criteria. I would suggest the authors to further revise this section accordingly. A revised manuscript in word-review mode would be useful to better and more immediately understand the real changes made by the authors in the manuscript, instead of the yellow highlighting.
- I would also suggest creating a specific subsection for the ethical statement. In addition to the IRB approval, I recommend including also a statement related to patients’ informed consent.
Reply: As requested a new paragraph, reporting the ethical statement, has been added and numbered as 4.1, consequently all the other paragraphs have been renumbered.
RR- The authors created this subsection. However, the authors should clearly state the study design and also disclose the exact study period.
- Moreover, the authors should clearly define the healthy volunteers/controls: it is not clear who they are, how/where exactly they were recruited, etc.
Reply: All the controls were age and sex matched volunteers recruited from the local healthy population. The demographic features of all controls are reported in table S2. A short sentence on the control group has been reported in paragraph 4.2.
RR- I would ask the authors to provide more precise information. It is not clear yet how these controls were recruited, if these are related to patients or not, how they were selected. Moreover, the information “All the controls were age and sex matched volunteers recruited from the local healthy population” is not included in the manuscript; however, as said, more information is needed to let understand the readership who these controls are and how they were effectively recruited, also to eliminate any concerns related to selection bias.
DISCUSSION
- There are many aspects deserving discussion. In the current version, this section sounds dispersive. Therefore, I suggest the authors start the discussion by schematically listing and highlighting their main findings. Moreover, I would recommend to re-organize the discussion in subsections as well or at least clearly highlight the specific discussion of each point that should be listed at the beginning of the introduction.
Reply: As suggested by this referee the discussion has been deeply reorganized. The first part schematically introduces the different tasks that will be further discussed. Following this scheme, the discussion has been divided into different subsections where each point introduced in the first part is discussed in detail.
RR- The authors tried to partially re-organize the discussion sections. They added a paragraph at the beginning of the discussion, which I suggest being further revised. They report their finding with the first general statement and then they repeat some methodological aspects, which were already highlighted at the end of the introduction. My recommendation was to clearly and schematically state the specific findings emerging from their results and, once “listed”, to discuss them one by one in the discussion. Indeed, I still perceive this section quite dispersive despite the partial re-organization made by the authors.
- there is no discussion regarding the study limitations. A specific paragraph should be added.
Reply: according to the referee’s suggestion a new subsection titled “ study limitation and future perspectives” has been added a the end of the discussion and numbered as 3.5.
RR- I think the study limitations section should be separated by the perspectives, which may better fit the conclusion. Moreover, a larger and specific discussion on the study limitations are needed, especially considering that here the authors clearly state that “most of our ALD patients were undergoing therapy at the time of saliva and blood sample collection, therefore we could not assess the potential impact of such treatments on the salivary proteome since we have limited data regarding untreated patients”. This is a very important limitation, which substantially limit the conclusion that can be made by the authors. Indeed, as mentioned above, the study design is not clear yet from the section method. Then, is this a cross-sectional study in which study participants were “randomly” in terms of disease course timing? If so, the general aim of the study (“Based on these considerations, the aim of the present study was to evidence by a top-down proteomic pipeline possible qualitative and/ or quantitative differences of targeted salivary proteins/peptides in patients with either AIH or PBC compared with healthy controls (HCs). may be significantly impaired”) because patients were not recruited at the diagnosis and, apparently, very few were. Therefore, the differences in the potential biomarkers may have definitely suffered from the previous clinical course and/or iatrogenic factors, instead of reflecting real immuno-pathologic differences between these diseases, also compared to controls.
RR- Therefore, some parts of the discussion and also the conclusion should be further revised (also in terms of strength) according to the study limitations, in my opinion.
Author Response
Overall, the authors clearly present the main concepts related to AIH, PBC, and PSC. St the beginning of the introduction, the authors correctly highlight that “Autoimmune liver diseases (AILDs) include a series of pathological conditions that target the liver and have a wide spectrum of presentation, ranging from asymptomatic forms to end stage liver disease requiring liver transplantation”. I think the authors (perhaps, after line 65) could also mention all those conditions and/or comorbidity which can be associated to an immune-mediated liver/biliary/cholecystic pathology, which can additionally complicate the differential diagnosis, such as celiac disease, for instance (see: Dig Liver Dis. 2016 Feb;48(2):112-9. doi: 10.1016/j.dld.2015.11.013; Nutrients. 2022 Oct 19;14(20):4379. doi: 10.3390/nu14204379.
Reply: as suggested by the referee we added in the introduction section a paragraph explaining the most common comorbidities linked to AIH and PBC. In particular, we insert a comment and a new reference focused on celiac disease.
RR- The authors added a paragraph. Revision accepted. However, the authors could highlight more the challenges for differential diagnosis in presence of comorbidities, especially if the related immunopathological processes may also affect other tract of the hepatobiliary system.
AR2- A sentence regarding differential diagnosis in presence of CEHAID and “overlap syndromes” has been added in the introduction section.
METHODS
- section 4.1: As mentioned above, the demographics should be included in the results, along with the description of clinical (also therapeutic) and laboratory aspects.
Reply: done
RR- Some demographic information is still present in this section, whereas numbers should be part of the result section only. This section should clearly and simply state the inclusion criteria used to identify the study population and, if any, exclusion criteria. I would suggest the authors to further revise this section accordingly. A revised manuscript in word-review mode would be useful to better and more immediately understand the real changes made by the authors in the manuscript, instead of the yellow highlighting.
AR2- as suggested all the demographic information are moved from methods to results. In this version the paragraph titled “study subjects and clinical studies” only describes criteria used to include or exclude patients and controls.
- I would also suggest creating a specific subsection for the ethical statement. In addition to the IRB approval, I recommend including also a statement related to patients’ informed consent.
Reply: As requested a new paragraph, reporting the ethical statement, has been added and numbered as 4.1, consequently all the other paragraphs have been renumbered.
RR- The authors created this subsection. However, the authors should clearly state the study design and also disclose the exact study period.
AR2- the paragraph “ethical statement” has been implemented with the study design. Salivary sampling has been performed during 2021 and this information has been included in the manuscript.
- Moreover, the authors should clearly define the healthy volunteers/controls: it is not clear who they are, how/where exactly they were recruited, etc.
Reply: All the controls were age and sex matched volunteers recruited from the local healthy population. The demographic features of all controls are reported in table S2. A short sentence on the control group has been reported in paragraph 4.2.
RR- I would ask the authors to provide more precise information. It is not clear yet how these controls were recruited, if these are related to patients or not, how they were selected. Moreover, the information “All the controls were age and sex matched volunteers recruited from the local healthy population” is not included in the manuscript; however, as said, more information is needed to let understand the readership who these controls are and how they were effectively recruited, also to eliminate any concerns related to selection bias.
AR2-the paragraph “study subjects and Clinical studies” has been implemented with inclusion and exclusion criteria applied to select healthy controls.
DISCUSSION
- There are many aspects deserving discussion. In the current version, this section sounds dispersive. Therefore, I suggest the authors start the discussion by schematically listing and highlighting their main findings. Moreover, I would recommend to re-organize the discussion in subsections as well or at least clearly highlight the specific discussion of each point that should be listed at the beginning of the introduction.
Reply: As suggested by this referee the discussion has been deeply reorganized. The first part schematically introduces the different tasks that will be further discussed. Following this scheme, the discussion has been divided into different subsections where each point introduced in the first part is discussed in detail.
RR- The authors tried to partially re-organize the discussion sections. They added a paragraph at the beginning of the discussion, which I suggest being further revised. They report their finding with the first general statement and then they repeat some methodological aspects, which were already highlighted at the end of the introduction. My recommendation was to clearly and schematically state the specific findings emerging from their results and, once “listed”, to discuss them one by one in the discussion. Indeed, I still perceive this section quite dispersive despite the partial re-organization made by the authors.
AR2: we apologize to the referee because we probably misunderstood its previous suggestions. The discussion has been reorganized consequently. However, we found this request mostly related to personal stylistic issues of the referee rather than conceptual ones. Two referees out of three found the discussion well organized and clear since from the first version.
- there is no discussion regarding the study limitations. A specific paragraph should be added.
Reply: according to the referee’s suggestion a new subsection titled “ study limitation and future perspectives” has been added a the end of the discussion and numbered as 3.5.
RR- I think the study limitations section should be separated by the perspectives, which may better fit the conclusion. Moreover, a larger and specific discussion on the study limitations are needed, especially considering that here the authors clearly state that “most of our ALD patients were undergoing therapy at the time of saliva and blood sample collection, therefore we could not assess the potential impact of such treatments on the salivary proteome since we have limited data regarding untreated patients”. This is a very important limitation, which substantially limit the conclusion that can be made by the authors. Indeed, as mentioned above, the study design is not clear yet from the section method. Then, is this a cross-sectional study in which study participants were “randomly” in terms of disease course timing? If so, the general aim of the study (“Based on these considerations, the aim of the present study was to evidence by a top-down proteomic pipeline possible qualitative and/ or quantitative differences of targeted salivary proteins/peptides in patients with either AIH or PBC compared with healthy controls (HCs). may be significantly impaired”) because patients were not recruited at the diagnosis and, apparently, very few were. Therefore, the differences in the potential biomarkers may have definitely suffered from the previous clinical course and/or iatrogenic factors, instead of reflecting real immuno-pathologic differences between these diseases, also compared to controls.
AR2: we acknowledge that our study has been performed on AIH and PBC patients that were heterogenous in terms of clinical parameters, grade of liver inflammation, stage of liver fibrosis and therapies, but recruitment of patients at diagnosis would have resulted in a study population even more heterogeneous, in fact: i) naïve patients are very difficult to recruit because most of the people suffering from autoimmune liver diseases arrive at diagnosis under pharmacological therapies due to the presence of comorbidities; ii) patients are heterogeneous at presentation with signs and symptoms of established chronic liver disease, up to decompensated cirrhosis; picture of acute, rarely fulminant, hepatitis with jaundice and marked elevation of transaminases serum levels; and rarely completely asymptomatic and seeking medical advice only because of liver laboratory abnormalities. To mitigate the bias due to differences in clinical presentation we decided to select patients only undergoing therapy since at last three years and therefore showing almost normal values of transaminases. Serum transaminases and IgG levels within the normal ranges usually define biochemical remission (Manns MP, Czaja AJ, Gorham JD, Krawitt EL, Mieli-Vergani G, Vergani D, et al. Diagnosis and management of autoimmune hepatitis. Hepatology 2010;51:2193-2213) and normal serum levels of AST and ALT for at least 2 years have been proposed as requisites before attempting treatment withdrawal also in subjects whit cirrhosis, which may have chronic elevation of the serum IgG level. (European Association for the Study of the Liver. EASL Clinical Practice Guidelines: autoimmune hepatitis. J Hepatol 2015;63:971-1004).
Based on these scientific evidences, we started a proteomic study with the aim of ”highlight by a top-down proteomic pipeline possible qualitative and/ or quantitative differences of targeted salivary proteins/peptides in patients with either AIH or PBC compared with healthy controls (HCs)” and to this purpose we selected patients with almost normal serum transaminase levels at the time of saliva sampling to reduce bias due to different clinical manifestation of the patients.
Furthermore, the laboratory data of AIH and PBC, and above all the transaminases, do not correlate at all with the salivary proteins which mostly discriminate the two groups selected by Boruta Algorithm. This indicates that salivary proteins are at least relatively independent of laboratory parameters measured in patients. As a result, despite the high heterogeneity of the clinical manifestation of the patients at time of saliva sampling, the robustness of the analytical and statistical approach used allowed to highlight not only a set of potential salivary biomarkers of AIH and PBC (t-test analysis) but also a panel of proteins able to discriminate the two groups of patients and classify them in the correct group (RF and LDA analyses). In our opinion, the results suggest that differences found in the salivary proteomic profile of ALD more likely reflect the real immuno-pathological differences between the two diseases, even with respect to controls, rather than the clinical course of the disease and/or iatrogenic factors.
RR- Therefore, some parts of the discussion and also the conclusion should be further revised (also in terms of strength) according to the study limitations, in my opinion.
AR2- discussion and conclusion have been revised to add considerations on the limitations linked to the transversal nature of the study and heterogeneity of the selected population.
Reviewer 3 Report
The authors satisfactorily addressed the raised points and the manuscript can be accepted.
Author Response
thank you for the work done,regards
Round 3
Reviewer 1 Report
The authors clarified several aspects. Of course, there are important limitations in the study design, which have been anyway disclosed and discussed. The control population is at least better defined in the revised version. The conclusion could be further improved, by providing more schematic and final messages, instead of focusing on study limitation; of course, the messages should be mitigated accordingly.
Author Response
We would like to thank the referee for its constructive contribution. Suggestions made by the referee have been used to improve our paper in a more comprehensive manner.
In the following our reply:
The authors clarified several aspects. Of course, there are important limitations in the study design, which have been anyway disclosed and discussed. The control population is at least better defined in the revised version. The conclusion could be further improved, by providing more schematic and final messages, instead of focusing on study limitation; of course, the messages should be mitigated accordingly.
AR: conclusion has been improved as follows:
The top-down proteomic pipeline exploited in this study allowed evidencing, for the first time, the qualitative and quantitative differences of targeted salivary proteins/peptides in AIH and PCB patients with respect to healthy controls .
Despite the high heterogeneity of the clinical manifestation of the patients the robustness of the analytical and statistical approach used allowed highlighting a set of potential salivary biomarkers of AIH and PBC. Biomarkers candidates of AIHp were individuated in peptides/proteins involved in antimicrobial defense, while peptides/proteins involved in innate immune system characterized PBCp.
RF analysis revealed the feasibility of the salivary proteome to discriminate groups of subjects based on AIH or PBC occurrence. On this regard RF, strengthened by LDA, appears an attractive machine-learning tool suited for classification of AIH and PBC based on their different salivary proteomic profile.
In our opinion, the results suggest that differences found in the salivary proteomic profile of AIH and PBC may reflect the immuno-pathological differences between the two diseases, even with respect to controls, rather than the clinical course of the disease and/or iatrogenic factors.